# Co-Suppression of *NbClpC1* and *NbClpC2*, Encoding Clp Protease Chaperons, Elicits Significant Changes in the Metabolic Profile of *Nicotiana benthamiana*

**DOI:** 10.3390/plants9020259

**Published:** 2020-02-18

**Authors:** Md. Sarafat Ali, Kwang-Hyun Baek

**Affiliations:** Department of Biotechnology, Yeungnam University, Gyeongsan, Gyeongbuk 38541, Korea; sarafatbiotech@ynu.ac.kr

**Keywords:** Clp protease, ClpC1, ClpC2, co-suppression, metabolites, virus-induced gene silencing

## Abstract

Metabolites in plants are the products of cellular metabolic processes, and their differential amount can be regarded as the final responses of plants to genetic, epigenetic, or environmental stresses. The Clp protease complex, composed of the chaperonic parts and degradation proteases, is the major degradation system for proteins in plastids. ClpC1 and ClpC2 are the two chaperonic proteins for the Clp protease complex and share more than 90% nucleotide and amino acid sequence similarities. In this study, we employed virus-induced gene silencing to simultaneously suppress the expression of ClpC1 and ClpC2 in *Nicotiana benthamiana* (NbClpC1/C2). The co-suppression of NbClpC1/C2 in *N. benthamiana* resulted in aberrant development, with severely chlorotic leaves and stunted growth. A comparison of the control and NbClpC1/C2 co-suppressed *N. benthamiana* metabolomes revealed a total of 152 metabolites identified by capillary electrophoresis time-of-flight mass spectrometry. The co-suppression of NbClpC1/C2 significantly altered the levels of metabolites in glycolysis, the tricarboxylic acid cycle, the pentose phosphate pathway, and the purine biosynthetic pathway, as well as polyamine and antioxidant metabolites. Our results show that the simultaneous suppression of ClpC1 and ClpC2 leads to aberrant morphological changes in chloroplasts and that these changes are related to changes in the contents of major metabolites acting in cellular metabolism and biosynthetic pathways.

## 1. Introduction

Plant proteases control a broad range of functions, including differentiation, development, cell death via gene expression, protein targeting, protein sorting, protein folding, protein quality control, and protein degradation [1]. Plastids have been assumed to originate from endosymbiotic ancestral cyanobacteria; therefore, to date, all characterized plastidal proteases are homologs of bacterial proteases [2,3]. In particular, chloroplast proteases are found in the envelopes, the stroma, the thylakoid lumen, and the thylakoid membranes. Chloroplasts contain three major types of proteases, namely FtsH, Deg, and Clp [4], that play important roles in the maintenance and biogenesis of chloroplasts. Both FtsH and Clp are ATP-dependent proteases, whereas Deg activity is independent of ATP. The operation sites in the chloroplasts can be differentiated as the stromal side where FtsH protease works and the luminal side where Deg protease works [5,6]. The Clp protease complex is localised in the stroma with some relation to the inner membrane [7]. 

Clp/Hsp100 proteins play diverse cellular functions as a class of molecular chaperones within the AAA+ (ATPases Associated with diverse cellular Activities) protein family. For example, most Clp/Hsp100 proteins work as the regulatory components of the Clp protease complex, which is the major machinery for the degradation of proteins in plastids [3,8,9]. As one of the most important enzymes for plant viability [10,11,12,13], the Clp protease system maintains chloroplast homeostasis [13,14,15,16] through the degradation of partially accumulated complexes or damaged proteins [2,17]. The proteolytic core of the *Arabidopsis thaliana* Clp protease complex is formed by two heptameric rings of plastome-encoded ClpP1 and nuclear-encoded ClpP3-P6 and ClpR1-R4 proteins stabilised by plant-specific ClpT1-T2 subunits. The Hsp100 chaperones (ClpC1-C2 and ClpD) unfold protein substrates for translocation to the proteolytic chamber [3,8,9]; some Clp degradation substrates can be directly recognised by these Hsp100 chaperones, while others are delivered for Clp-mediated degradation by a binary adaptor system formed by ClpS and ClpF proteins [8,18]. The various Clp subunits exert distinct functional differences in plant growth and development [13,19]. To date, all *Arabidopsis* Clp subunits have been characterized in both green and non-green plastids [4,13,20]. 

ClpC1 and ClpC2 contain many similarities at both the nucleotide and amino acid sequence levels and are key chaperones for the Clp protease systems in chloroplasts [21,22]. In addition to a role in Clp proteolytic activity, ClpC proteins also import cytosolic preproteins into the chloroplast in association with the integral membrane proteins Tic110, Tic40, Tic20, and the intermembrane space protein Tic22, as well as other stromal chaperones (cpHsp70, Hsp90C) [7,23,24,25,26]. In tobacco, when ClpC1 and ClpC2 genes were simultaneously suppressed using the antisense technique, plants failed to produce viable cell lines [27]. *Arabidopsis* ClpC1 null mutants exhibit significant phenotypic changes, the most prominent of which are slower growth rates, leaf chlorosis, and impaired photosynthetic activity, whereas ClpC2 null mutants show a wild-type like phenotype [21,28,29]. The overexpression of ClpC2 can fully complement the loss of ClpC1 in *Arabidopsis* [11,16], suggesting that both perform similar, if not identical, functions in the chloroplast. ClpC1 is required for the degradation of deoxyxylulose-5-phosphate synthase (DXS) in the methylerythritol-4-phosphate (MEP) pathway [30]. Moreover, *Arabidopsis* ClpC1 knockout mutants present a decrease in the efficiency of degradation and import of proteins in chloroplasts [31].

Metabolomics is the study of the metabolic profile of molecules in a biological system that enables the analysis of cellular functions via a holistic view of metabolite pathways, and the set of metabolites synthesised by an organism constitutes its metabolome [32]. A systematic analysis of metabolites can provide a solid and valid proof for a quantitative rather than qualitative description of cellular regulation [33]. A biological system can be analyzed in the aspect of the metabolome as a ‘link between genotype and phenotype’ due the metabolism being an integrated state of a genetic response to the environmental factors [34]. By comparing the metabolomes of samples, insights can be gained into the genetic, environmental, and developmental modulators that distinguish the samples. Of several metabolite measurement techniques, Capillary Electrophoresis Time-Of-Flight Mass Spectrometry (CE-TOF-MS) can be used for the simultaneous profiling of energy metabolic pathways, e.g., glycolysis, the tricarboxylic acid (TCA) cycle, and the amino acid and nucleotide biosynthetic pathways [35]. 

Gene silencing in *N. benthamiana*, later confirmed by semi-quantitative RT-PCR, was achieved through tobacco rattle virus (TRV)-based VIGS as described in previous reports [36,37,38,39,40]. To date, several studies have reported the functional roles of ClpC1 and ClpC2 proteins [7,23,24,25,26], but the chaperonic roles of the Clp protease complex in relation to cellular metabolism remain unknown. Additionally, mutant analyses have relied on single ClpC1 or ClpC2 mutants due to double mutant-associated lethality. Therefore, in the present study, we investigated the effects of suppressing both *Nicotiana benthamiana* ClpC1 and ClpC2 (NbClpC1 and NbClpC2, NbClpC1/C2) on metabolite levels. Employing virus-induced gene silencing (VIGS), we simultaneously suppressed NbClpC1/C2 in *N. benthamiana* plants using only one silencing vector, and subsequently identified their roles in metabolite pool changes.

## 2. Results

### 2.1. Metabolites of the NbClpC1/C2 Co-Suppressed Leaves 

Using CE-TOF-MS as described in the Methods section, we analysed the metabolite profiles of the control and NbClpC1/C2 co-suppressed *N. benthamiana* leaves. A total of 152 metabolites (106 in cation mode and 46 in anion mode) were detected based on the Human Metabolome Technologies (HMT) Inc. metabolite database (Appendix A). The rate of relative area between the control and the NbClpC1/C2 co-suppressed *N. benthamiana* leaves was calculated for 152 putative metabolites. Welch’s *t*-test was performed between the groups. Out of 152 metabolites, 64 metabolites had significantly changed endogenous levels in the suppressing lines. Owing to the complexity of the acquired data, as reflected in the complexity of the spectral data, statistical multivariate analyses (e.g., hierarchical cluster analysis (HCA)) were performed to ensure analytical rigorousness and define both the similarities and differences among the samples (Figure 1). A heat map of the HCA results (Figure 1) shows that co-suppression of NbClpC1/C2 resulted in significant changes in the metabolic profile. The metabolites showing high levels in the control plants presented low levels in the NbClpC1/C2 co-suppressed plants, and vice-versa. 

### 2.2. Effects of NbClpC1/C2 Co-Suppression on the Content of Intermediate Metabolites of Glycolysis

The co-suppression of NbClpC1/C2 in *N. benthamiana* significantly reduced the concentrations of all intracellular metabolites in glycolysis (Figure 2). With the co-suppression of NbClpC1/C2, the glucose-6-phosphate (G6P) content was only 8.1% and that of fructose-6-phosphate (F6P) only 10.1%, that of control plants. Dihydroxyacetone phosphate (DHAP), 3-phosphoglycerate (3PG), 2-phosphoglycerate (2PG), and pyruvate contents were below the detection level, while the level of lactate, an end product of glycolysis, was not significantly different between the control and the NbClpC1/C2 co-suppressed *N. benthamiana* plants.

### 2.3. Effects of NbClpC1/C2 Co-Suppression on the Levels of Intermediate Metabolites of the TCA Cycle 

The co-suppression of NbClpC1/C2 in *N. benthamiana* markedly altered the levels of intermediate metabolites in the TCA cycle (Figure 3). With the co-suppression of NbClpC1/C2, the citrate and succinate contents increased 17.0- and 4.2-fold, respectively, compared to the control plants, whereas the fumarate and malate contents decreased to 60.0% and 19.7% of those in the control, respectively (Figure 3). Cis-aconitate was below the detectable limit in the control plants.

### 2.4. Effects of NbClpC1/C2 Co-Suppression on the Levels of Intermediate Metabolites of the Pentose Phosphate Pathway 

The levels of intermediate metabolites in the pentose phosphate pathway in *N. benthamiana* leaves were also affected by the co-suppression of NbClpC1/C2 (Figure 4). The glucose-6-phosphate (G6P), sedoheptulose-7-phosphate (S7P), and fructose-6-phosphate (F6P) contents in NbClpC1/C2 co-suppressed *N. benthamiana* plants were significantly lower than those in the controls, presenting values of 8.1%, 52.6%, and 10.1%, respectively.

### 2.5. Effects of NbClpC1/C2 Co-Suppression on Polyamine and Antioxidant Contents 

Changes in polyamine and antioxidant metabolite contents were also found with the co-suppression of NbClpC1/C2 in *N. benthamiana* (Figure 5). The co-suppression of NbClpC1/C2 significantly increased the spermine, cysteine, and oxidised glutathione (GSSG) contents by 2.4-, 3.9-, and 2.0-fold, respectively. The co-suppression of NbClpC1/C2 also significantly reduced the S-adenosyl methionine (SAM) content, which was only 46.6% of the control plants; in contrast, ornithine, putrescine, spermidine, and reduced glutathione (GSH) contents were not significantly different.

### 2.6. Effects of NbClpC1/C2 Co-Suppression on Purine Nucleotide Levels 

The co-suppression of NbClpC1/C2 in *N. benthamiana* also altered the contents of intermediate metabolites in the purine biosynthetic pathway (Figure 6). In NbClpC1/C2 co-suppressed plants, the adenosine and guanosine levels increased 2.5- and 3.7-fold, respectively, compared to those in the controls. Guanine and ATP levels also increased with the co-suppression of NbClpC1/C2, whereas those of ADP were not significantly different.

## 3. Discussion

Vascular plants contain two closely related ClpC genes that encode chaperonic ClpC1 and ClpC2 that share high sequence similarities [22,41,42]. They are also functionally similar, as indicated by the ability of overexpressed ClpC2 to complement the chlorotic phenotype of the *Arabidopsis* ClpC1 mutant [11,16]. In *Arabidopsis*, the ClpC1 and ClpC2 double homozygous mutant is embryonically lethal [21], making it impossible to use this model plant to investigate the chaperonic roles of ClpC1 and ClpC2 in plant development and physiology. Therefore, to understand the chaperonic roles of the Clp protease complex, we performed VIGS in *N. benthamiana* to simultaneously suppress NbClpC1 and NbClpC2 using only one silencing vector.

The co-suppression of NbClpC1/C2 resulted in severe leaf-yellowing and growth retardation phenotypes, indicating that reduced NbClpC1/C2 activity leads to abnormalities at the subcellular level [43]. The NbClpC1/C2 co-suppressed plants also produced more branches with broader, thicker, and wider lower leaves, while the upper leaves were smaller and showed pleiotropic phenotypes [43]. The chlorotic appearance of the NbClpC1/C2 co-suppressed plants was maintained throughout vegetative growth and even though the plants flowered, they failed to produce seeds. 

The significant reduction in the levels of all the metabolites of glycolysis indicates a lower accumulation of intermediates in this pathway (Figure 2). As the main function of glycolysis is to generate high-energy molecules, pyruvate, and intermediate compounds that serve as important starter materials for other cellular processes, the significant reduction in metabolite levels in the glycolytic pathway may be related to the disrupted production of high-energy molecules, pyruvate, and other intermediate products. The co-suppression of NbClpC1/C2 significantly reduced the photosynthetic capacity in *N. benthamiana* [44], suggesting that reduced levels of photosynthates resulting from a lowered photosynthetic capacity can affect the glycolytic pathway. 

Glycolysis is the main pathway for the oxidation of carbohydrates generated during photosynthetic ‘carbon fixation’, followed by the TCA cycle. The TCA cycle is a central metabolic pathway for aerobic processes and is responsible for most of the carbohydrate, fatty acid, and amino acid oxidation, a process that produces energy and reducesing power [45]. With co-suppression of NbClpC1/C2, the levels of most intermediates of the TCA cycle were significantly increased at the beginning of the cycle, and then decreased gradually compared to the control (Figure 3). Phosphofructokinase-1 (PFK-1) as the key regulatory step in glycolysis converts fructose 6-phosphate to fructose 1,6-bisphosphate. PFK-1 is considered as a metabolic valve that regulates the rate of glycolysis. PFK-1 is regulated by one of its downstream products, citrate, which acts as an allosteric inhibitor of the PFK-1, slowing its activity [46]. The high levels of citrate in NbClpC1/C2 co-suppressed plants indicate that it is not being consumed by the TCA cycle and glucose breakdown might be slowed due to the regulatory effect of citrate on PFK-1. As citrate plays an important role in the coordination between glycolysis and the TCA cycle through controlling glycolysis, a significant increase of citrate might illustrate the occurrence of imbalanced products in glycolysis and the TCA cycle.

The TCA cycle of plants has unique features as it contains a malic enzyme and phosphoenolpyruvate (PEP) carboxylase [47]. The malic enzyme and PEP carboxylase provide plants with metabolic flexibility for the metabolism of malate and PEP, respectively. The malic enzyme in the mitochondrial matrix converts malate into pyruvate, and PEP carboxylase in the cytosol converts PEP to malate. According to the bottom-up regulation of plant respiration, a high citrate content might inhibit pyruvate and malate synthesis from PEP [47].

Cellular respiration is also inhibited by its products and the intracellular level of ATP is a key regulator of cellular respiration. When the levels of ATP are high in a cell, the cell has a high amount of free energy. Due to this high amount of free energy, the cellular pathways that generate ATP are slowed, or downregulated. In NbClpC1/C2 co-suppressed plants, the contents of ATP significantly increased as compared to the control plants (Figure 6). The increased contents of ATP might inhibit the oxidative phosphorylation, TCA cycle, pyruvate oxidation, and glycolysis [46]. Therefore, the bottom-up control of the respiratory carbon pathways in NbClpC1/C2 co-suppressed plants might play an important role in adjusting the demand for biosynthetic building blocks. 

The pentose phosphate pathway is an alternative oxidative pathway involving interconversions and rearrangements of sugar phosphates, that results in the net production of NADPH in the absence of photosynthesis. The starting material, glucose-6-phosphate, undergoes different reactions depending on whether there is a greater need for ribose-5-phosphate or NADPH. As the pentose phosphate pathway generates reducing equivalents (i.e., NADPH), as well as ribulose-5-phosphate and erythrose-4-phosphate, reduced metabolite levels will likely result in decreased production of NADPH and other riboses (Figure 4).

Oxidative stress leads to profound alterations in various biological structures, including cellular membranes, lipids, proteins, and nucleic acids. Ornithine, putrescine, SAM, and spermidine serve as source materials for spermine synthesis [48]. Therefore, slightly decreased putrescine and spermidine and significantly decreased SAM contents indicate that they may have been used for spermine synthesis or that spermine may not have been efficiently catalysed. Ethylene is synthesised from SAM [49] and the reduced SAM content might have affected ethylene synthesis. We observed that NbClpC1/C2-suppressed plants completed their life cycle four weeks later than the control plants. As ethylene promotes senescence, ethylene production may have been hampered due to the reduced SAM content, thereby increasing the life cycle of the NbCpC1/C2 co-suppressed plants. Consequently, the marked changes in the contents of polyamines and antioxidant molecules are consistent with the dwarf phenotype and the longer life cycle of the NbClpC1/C2 co-suppressed plants. 

GSH is used by glutathione peroxidase to detoxify H_2_O_2_. In the detoxification reaction, the oxidation potential of hydrogen peroxide is transferred to GSSG [50], which is subsequently recycled by glutathione reductase using NADPH. The regenerated GSH is then ready to detoxify more H_2_O_2_. When the antioxidant defence system is functioning properly, most of its components are present as reduced forms of GSH and NADPH. If there is an impairment, then the oxidised components (H_2_O_2_, GSSG, and NADP^+^) tend to accumulate, with adverse effects. The entire process is dependent on the energy production at the cellular level, healthy mitochondrial function, and an active pentose phosphate metabolic pathway, with the latter being especially important for providing NADPH. High GSSG levels indicate that a cell has suffered from oxidative stress; for example, GSSG levels are significantly higher in transgenic seedlings during periods of stress [51,52], therefore, our increased GSSG levels in the NbClpC1/C2 co-suppressed plants indicate the co-suppression significantly induced severe stresses to the plants, especially oxidative stress.

Purine and pyrimidine nucleotides are major energy carriers, as well as subunits of nucleic acids and precursors for the synthesis of nucleotide cofactors such as NAD [53]. In the NbClpC1/C2 co-suppressed plants, the levels of most energy carrier molecules increased for recovery from the chlorotic phenotype. As ClpC1 and ClpC2 proteins form the ATPase subunit of the Clp protease complex, ATP may be misprocessed in NbClpC1/C2 co-suppressed *N. benthamiana* plants. Thus, the increased ATP content observed in the NbClpC1/C2 co-suppressed *N. benthamiana* plants is consistent with the NbClpC1/C2 co-suppression (Figure 6).

The metabolomics analysis revealed that the co-suppression of NbClpC1/C2 in *N. benthamiana* plants slows glycolysis, the TCA cycle, and the pentose phosphate pathway, leading to significant increases in the levels of polyamines, antioxidant metabolites, and energy carrier molecules. In conclusion, co-suppression of NbClpC1 and NbClpC2 in *N. benthamiana* results in severe chlorosis, aberrant development, and the semi-dwarf phenotype, effects that are accompanied by significant changes in the contents of various metabolites.

## 4. Materials and Methods

### 4.1. Plant Materials, VIGS, and Semi-Quantitative RT-PCR

Seeds of *N. benthamiana* were sown and grown in plastic pots (12 cm diameter × 10 cm height) in a mixture of 70% coco peat, 17% peat moss, 5% zeolite, and 8% perlite. The plants were regularly watered and grown under fluorescent lights at 120 μEinstein m^−2^ s^−1^ under a 16/8 h light/dark regime at 22 ± 2 °C in a controlled, walk-in chamber. VIGS vector construction, infiltration of *Agrobacterium* into leaves of *N. benthamiana* plants, and confirmation of co-suppression have been described in detail in our previous research [43] and in the Appendix A (Appendix A). In this research, we silenced 12 plants with TRV2:GFP vector (control), and 12 plants with TRV2:NbClpC vector, respectively. Metabolite profiling was performed using the sixth leaves above the infiltrated leaves after 4 weeks of infiltration.

### 4.2. Preparation of Samples for Metabolic Profiling 

The sixth leaves above the *Agrobacterium*-infiltrated leaves were used for metabolic profiling. The fresh weight of the leaves was used for metabolite analysis. The sample preparation was performed by pooling the sixth leaves from six plants for each treatment. Frozen samples were transferred into a cryotube containing 500 µL of methanol mixed with 50 µm of an external standard (methionine sulfone for cation analysis and d-camphor-10-sulfonic acid for anion analysis) and frozen in liquid nitrogen. After homogenisation with a multi-beads shocker (MS-100R; Tomy, Tokyo, Japan) at 4000 × *g* for 60 s at 4 °C, 500 µL of chloroform and 200 µL of Milli-Q water were added to the homogenate, mixed well, and centrifuged at 2300× *g* for 5 min at 4 °C. The resultant water phases were ultrafiltrated using the Millipore Ultrafree-MC Centrifugal 5 kDa filter unit (Millipore, Billerica, MA, USA). The filtrate was lyophilised, dissolved in 50 μL of Milli-Q water, and analysed by CE-MS.

### 4.3. CE-TOF-MS Analysis 

The CE-MS analysis was performed using an Agilent CE system equipped with a time-of-flight mass spectrometer (TOF-MS) (Agilent Technologies, USA) as previously described [54,55,56,57]. Metabolites in the samples were identified by comparing the migration times (MTs) and *m/z* ratios with those of authentic standards. The tolerance was ±0.5 min for MTs and ±10 ppm for the *m/z* ratios. Metabolites in the samples were quantified by comparing their peak areas with those of the authentic standards using ChemStation software (Agilent Technologies). Data analyses were conducted using GraphPad Prism. The variance of sample data was analysed by ANOVA using Statistical Analysis Software (SAS) version 9.4 (SAS Inc., Cary, NC, USA). 

## Figures and Tables

**Figure 1 plants-09-00259-f001:**
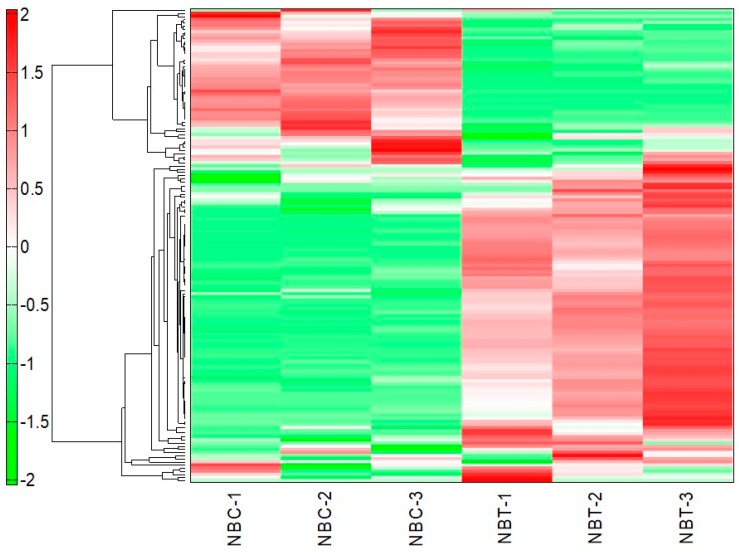
Heat map of the results of the hierarchical cluster analysis of metabolites in the leaves of NbClpC1/C2 co-suppressed *Nicotiana benthamiana.* The horizontal and vertical axes show sample names and peaks, respectively. The distances between samples and between peaks are displayed in three diagrams. NBC-1, 2, 3: Control 1, 2, and 3; NBT-1, 2, 3: NbClpC1/C2 co-suppressed *N. benthamiana* 1, 2 and 3.

**Figure 2 plants-09-00259-f002:**
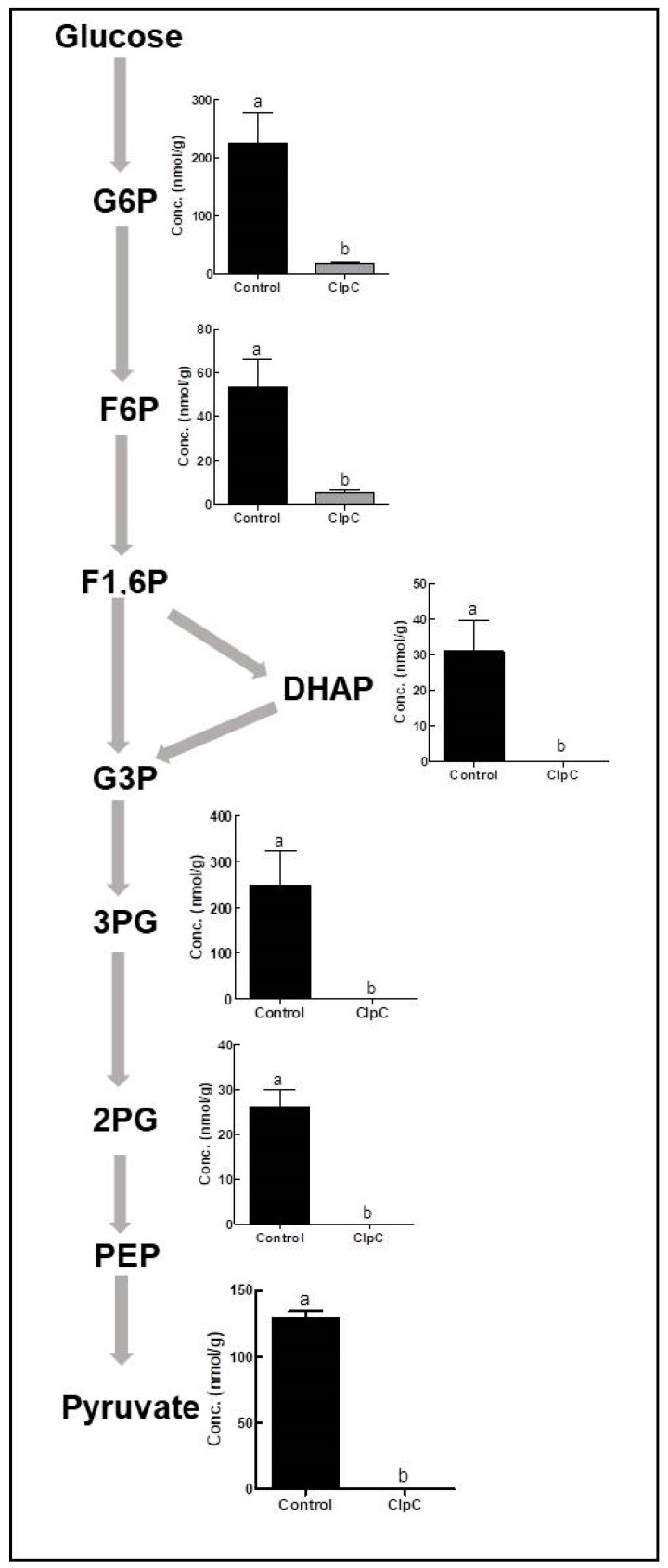
Comparison of glycolysis metabolite contents between control and NbClpC1/C2 co-suppressed *Nicotiana benthamiana* plants. G6P, glucose-6-phosphate; F6P, fructose-6-phosphate; F1,6P, fructose-1,6-bisphosphate; G3P, glyceraldehyde-3-phosphate; DHAP, dihydroxyacetone phosphate; 3PG, 3-phosphoglyceric acid; 2PG, 2-phosphoglyceric acid; PEP, phosphoenolpyruvate. The error bars indicate the standard deviation (SD) of triplicate samples. The different letters on the bars indicate statistically significant differences between groups (*p* < 0.05).

**Figure 3 plants-09-00259-f003:**
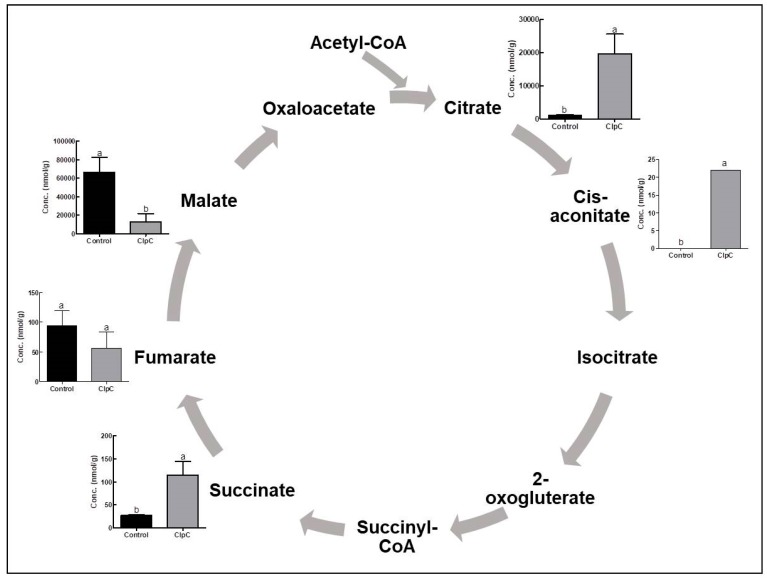
Changes in TCA cycle-related metabolite contents with co-suppression of NbClpC1/C2. ClpC: NbClpC1/C2 co-suppressed *N. benthamiana*. The error bars indicate the SD of triplicate samples. The different letters on the bars indicate statistically significant differences between groups (*p* < 0.05).

**Figure 4 plants-09-00259-f004:**
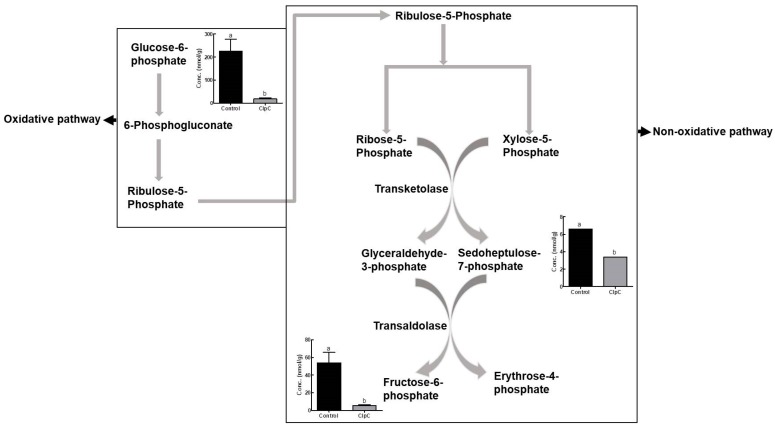
Comparison of pentose phosphate pathway-related (oxidative and nonoxidative) metabolite contents in *Nicotiana benthamiana* plants with co-suppression of NbClpC1/C2. ClpC: NbClpC1/C2 co-suppressed plants. The error bars indicate the SD of triplicate samples. The different letters on the bars indicate statistically significant differences between groups (*p* < 0.05).

**Figure 5 plants-09-00259-f005:**
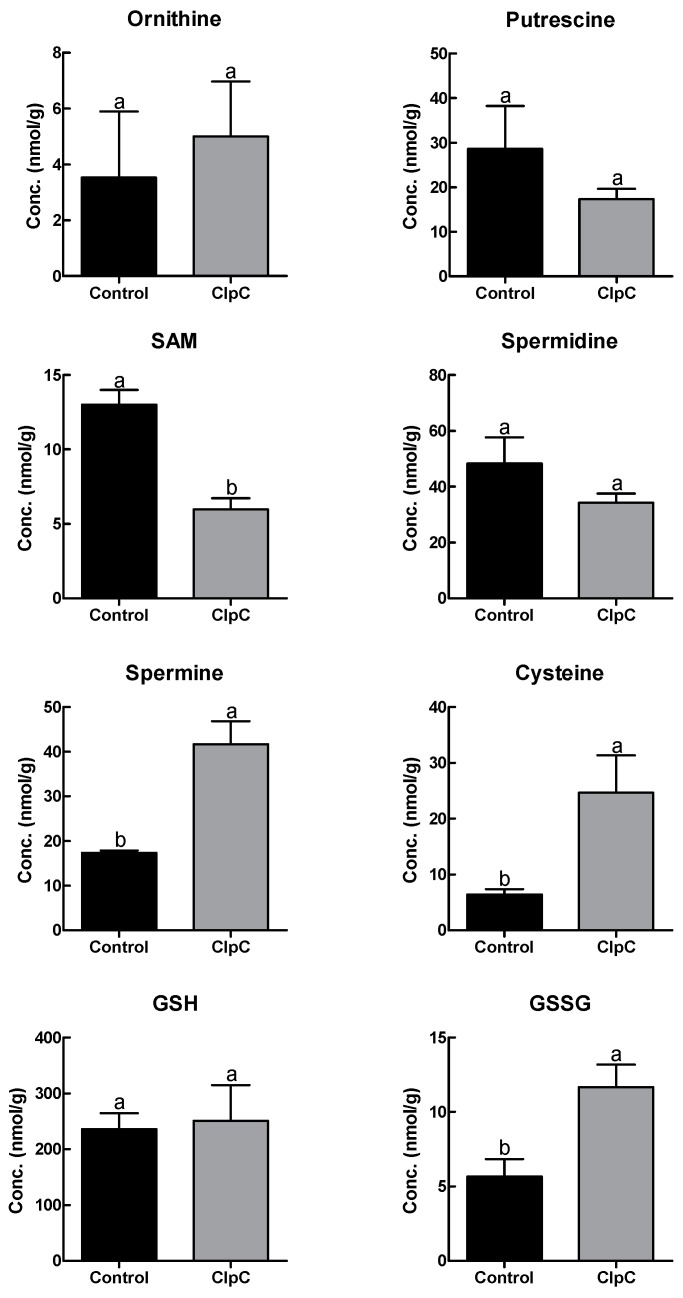
Changes in polyamine and antioxidant metabolite contents with co-suppression of NbClpC1/C2 in *Nicotiana benthamiana* plants. ClpC: NbClpC1/C2 co-suppressed plants. SAM, S-adenosyl methionine; GSH, reduced glutathione; GSSG, oxidised glutathione. The error bars indicate the SD of triplicate samples. The different letters on the bars indicate statistically significant differences between groups (*p* < 0.05).

**Figure 6 plants-09-00259-f006:**
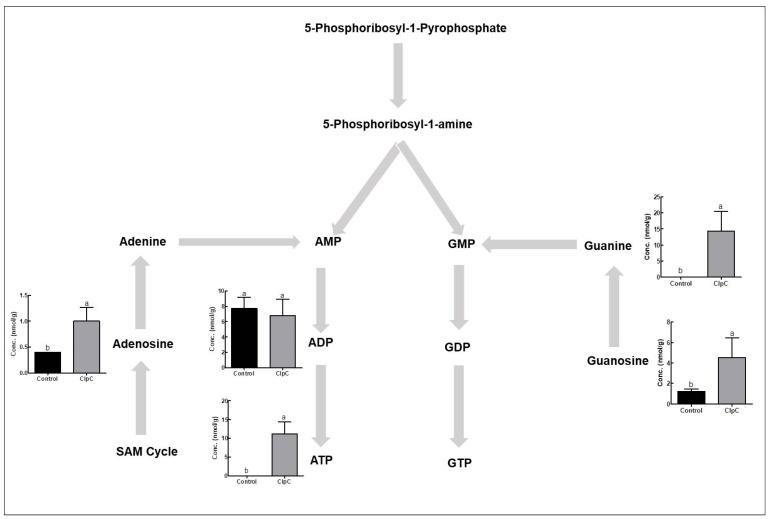
Changes in purine nucleotide biosynthetic pathway-related metabolite contents in *Nicotiana benthamiana* plants with co-suppression of NbClpC1/C2. ClpC: NbClpC1/C2 co-suppressed plants. The error bars indicate the SD of triplicate samples. The different letters on the bars indicate statistically significant differences between groups (*p* < 0.05).

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
