# Peer review of "Co-Suppression of NbClpC1 and NbClpC2, Encoding Clp Protease Chaperons, Elicits Significant Changes in the Metabolic Profile of Nicotiana benthamiana"

_plants, 2020, doi:10.3390/plants9020259_

Round 1
Reviewer 1 Report
The manuscript has addressed metabolome analysis in NbClpC1 and 2 suppressed Nicotinana benthamiana. NbClpC1 and 2 is components of Clp protease complex, which is major degradation system for proteins in plastids. Metabolome analysis shows that suppression of NbClpC1 and 2 affects several metabolic pathways, such as glycolysis, TCA cylcle, pentose phosphate pathway and purine nuleotide biosynthetic pathway. This manuscript provides the useful information to reveal the functions of NbClpC1 and 2. However, there are concerns to address before publication.
There is no overview of metabolome analysis in "Result" section. Before clustering analysis, the manuscript should describes how many metabolites significantly changes their endogenous levels in suppressing lines. Moreover, these significantly changed metabolites should be mapped in biosynthetic pathways. And then, biosynthetic pathways affected by suppression of NbClpCs should be extracted. Method section describes that semi-quantitative RT-PCR has performed to confirm the suppression of NbClpCs. However, there is no description in "Results" section. In lines 179 and 180, it is described that "this confirmation has been achieved in previous reports". These previous reports should be referred in "Introduction". The readers will not realize that NbClpCs are successfully suppressed until they read "Discussion".Author Response
Please see the attachment.

Reviewer 2 Report
The manuscript entitled “Co-suppression of NbClpC1 and NbClpC2, encoding Clp protease chaperons, elicits significant changes in the metabolic profile of Nicotiana benthamiana” describes the metabolic changes in N. benthamiana plats after the silencing of NbClpC1 and NbClpC2 using VIGS. The manuscript is well written and easy to read.
The introduction and discussion clear and sufficient. As a general comments, authors could extend the description of the general changes induced by the silencing, For example: authors describe in line 94 that 152 putative metabolites were calculated, all the calculated metabolites show significant differences between GFP-silenced and NbClpC1/C2 co‐suppressed plants? In the same way, there is any other pathway altered besides the ones described in points 2.2 to 2.6. As a Specific comment, all the graphs showed in the figures are described in the results section except for the cis-aconitate.
On the materials and methods section, authors describe that “confirmation of co‐suppression have been described in detail in our previous research” Please consider to add a figure (maybe as supplementary material) to illustrate the level of suppression reached.
Reviewer 3 Report
Dear Authors,
Please make sure the following issues are solved.
You check the statistical significance of many compounds in a large dataset. This can result in a large number of false discoveries if used with the usual threshold p < 0.5. Therefore, adjustment of the p-value thresholds is necessary so that the overall error rate of the findings in the paper is set p < 0.05. Consider using Bonferroni correction or similar (Fig 2 and elsewhere). Several of your "phenomena" will turn to false positives (Supplementary tables), but this is a must. N.d. values can be 0's in statistical models, however.
Are your metabolite data given on dry or fresh weight basis? General phenotypic changes are likely to result in disturbed water balance, making the dry weight a default choice.
CE has known retention time reproducibility issues compared to HPLC-MS or UPLC-MS. Nevertheless, the given 0.5 min tolerance for feature retention time variation seems a bit high to me. What measures were taken to avoid false identification? Was integration semi-automatic? Please add a series of electropherograms to the S. Material.
The provided references for CE-MS [54-57] were not the publications of the current authors, therefore I do not think the term "as previously described" can be used. Please revise and provide detailed CE-MS instrument and measurement parameters (interface, capillary type; BGE composition, injection type and time, voltage profile, etc.).
I do not think the control should be referred to as "GFP-silenced" (figures and elsewhere) - there is no GFP gene in the plant used, is it? Consider using simply "control" instead and explain in 4.1.
English: L38: proteases function on the stromal side ... (?)
Best regards.
Round 2
Reviewer 1 Report
The revised manuscript have been addressed the issues raised in previous review. The manuscript is now acceptable for publication.